# Is Brain Network Efficiency Reduced in Young Survivors of Acute Lymphoblastic Leukemia?—Evidence from Individual-Based Morphological Brain Network Analysis

**DOI:** 10.3390/jcm11185362

**Published:** 2022-09-13

**Authors:** Ying Zhuang, Long Qian, Lin Wu, Linglong Chen, Fei He, Shouhua Zhang, Gerald L. Cheung, Fuqing Zhou, Honghan Gong

**Affiliations:** 1Department of Oncology, Jiangxi Province Hospital of Integrated Chinese and Western Medicine, No. 90, Bayi Road, Xihu District, Nanchang 330003, China; 2Department of Biomedical Engineering, College of Engineering, Peking University, No. 60 Yannan Yuan, Beijing 100871, China; 3Department of Radiology, The First Affiliated Hospital of Nanchang University, Yongwaizheng St. 17, Nanchang 330006, China; 4Department of Hematology, Jiangxi Provincial Children’s Hospital, Diezihu Ave. 1666, Nanchang 330000, China; 5Spin Imaging Technology Co., Ltd., No.6 Fengxin Road, Nanjing 210012, China; 6Neuroradiology Laboratory, Jiangxi Province Medical Imaging Research Institute, Yongwaizheng St. 17, Nanchang 330006, China

**Keywords:** gray matter morphological networks, acute lymphoblastic leukemia, young survivors, executive efficiency

## Abstract

Altered cerebral structure and function have been observed in young survivors of acute lymphoblastic leukemia (ALL). However, the topological organization of the morphological brain networks (MBNs) has not yet been investigated at the individual level. Twenty-three young survivors of ALL and twenty healthy controls (HCs) were recruited and underwent T1-weighted magnetic resonance imaging (MRI) scanning. After preprocessing and segmentation, individual-based MBNs were constructed based on the morphological similarity of gray matter using the combined Euclidean distance. Young survivors showed a significantly lower global clustering coefficient (*p* = 0.008) and local efficiency (*p* = 0.035) compared with HCs. In addition, ALL survivors exhibited bidirectional alterations (decreases and increases) in nodal centrality and efficiency around the Rolandic operculum and posterior occipital lobe (*p* < 0.05, false discovery rate (FDR) corrected). Altered nodal topological efficiencies were associated with off-therapy duration and verbal memory capacity in the digit span test (*p* < 0.05, FDR corrected). Network-based statistical analysis revealed decreased morphological connections mainly in the pallidum subnetwork, which was negatively correlated with off-therapy durations (*p* < 0.05). Overall, the topological organization of the individual-based MBNs was disrupted in the young survivors of ALL, which may play a crucial role in executive efficiency deficits.

## 1. Introduction

With advances in treatment, the 5-year event-free survival rates in pediatric patients with acute lymphoblastic leukemia (ALL) now vary from 76% to 86% and even exceed 90% in some reports [1]. In addition to the interruption of education due to illness, neurotoxic drugs are administered that cross the blood–brain barrier and, through neuronal apoptosis, inflammation, and inhibition of hippocampal neurogenesis, result in white matter (WM) deficiencies and disruptions in the myelination process that ultimately impact long-term functional outcomes and quality of life in ALL survivors [2]. Neurocognitive impairment is the most common consequence, with higher than expected rates among young survivors of ALL, including 13% of patients experiencing memory impairment, 17–54% with executive dysfunction, and 62% with attention deficits and difficulties in task execution [2].

Recently, neuroimaging studies have provided structural and functional evidence for neurocognitive impairments in young survivors of ALL. For example, a smaller hippocampal volume or surface area [3,4], impaired microstructural WM integrity in frontal brain regions [5], decreased functional connectivity density [6], and reduced functional network efficiency [7] are closely related to eventual cognitive impairment [3,4,5,6,7]. Comparatively, the findings with structural WM networks and functional correlation networks could better explain the cognitive symptoms of the patients [6,7,8,9]. This information suggests that an analysis of topological properties at the network level is more conducive to revealing the central substrates of neurocognitive impairments in young survivors of ALL.

However, little is known about the topological properties of gray matter (GM) morphological brain networks (MBNs) in young survivors of ALL. Past investigations of most morphological brain networks required population-based studies [10], which limit their application at the individual level and in clinical practice. To date, several methods have been proposed to construct individual-based MBNs, providing a robust and valuable tool to better understand brain abnormalities [11,12].

Given previous evidence of decreased resting-state functional connectivity density [6], we hypothesized that (i) young survivors of ALL would show disrupted global and nodal topological organization in their GM morphological networks and that (ii) the discovered abnormal morphological connectivity or topological properties would be associated with clinical characteristics. Unlike measures of GM volume or cortical thickness, individual-based morphological networks may allow for the sensitive detection of subtle brain maldevelopment with structural complexity under pathological conditions [12,13]. Here, we constructed single-subject GM networks based on multiple morphological features and multivariate Euclidean distance [14] and applied graph theoretical analysis to investigate the topology of connections in young survivors of ALL. These findings might provide valuable information for morphological insights into the chemotherapy-induced, long-term effects of brain changes in pediatric ALL patients.

## 2. Materials and Methods

### 2.1. Participants

This study included 25 children with a history of ALL who were off-therapy for at least 12 months at the time of enrollment and 22 healthy children as healthy controls (HCs). The young survivors of ALL were recruited from the outpatient clinic of the First Affiliated Hospital of Nanchang University and Jiangxi Provincial Children’s Hospital. All patients were treated with a standardized three-year treatment using the Chinese Childhood Leukemia Group (CCLG)-ALL 2008 protocol [15]. Young survivors of ALL were excluded based on a history of cranial radiation, gross neuropathologies (e.g., leukomalacia or ventriculomegaly), major sensory impairments, or any significant medical or psychiatric condition diagnosed before or unrelated to ALL. The age at onset was between 6 and 14 years in participants with ALL. All of the HCs were recruited from the local area through community postings and screened using the Clinical Diagnostic Interview Nonpatient Version. HCs had no current or lifetime diagnosis of significant cognitive disorders, head trauma, or magnetic resonance imaging (MRI) contraindications.

The study was approved by the Medical Research Ethics Committee and the Institutional Review Board of the First Affiliated Hospital of Nanchang University (Protocol number 2018039). Written informed consent was obtained from the parents or legal guardians of each participant because our participants were under the age of 18 years.

### 2.2. Clinical Assessment

Clinical assessments were performed by two experienced psychiatrists and included the following measures: (i) full scale of the Wechsler Adult Intelligence Scale, Fourth Edition [4,6], to assess intelligence quotient (IQ; average IQ is 100); (ii) Trail Making Test part A (TMT-A; average is 29 s), to assess central executive functioning with a deficient score being greater than 78 s; and (iii) Wechsler Memory Scale Digit Span Test (Forward) (WMS-DST), to assess verbal memory capacity for short-term memory, with scores > 6–7 being normal for adolescents.

### 2.3. MRI Data Acquisition

All the participants were scanned on a 3-T MRI system (Siemens Trio, Munich, Germany) with the same sequence and an 8-channel head coil. A high-resolution magnetization-prepared rapid gradient echo (MPRAGE) T1-weighted sequence was used for morphological MRI data acquisition with the following parameters: repetition time = 1900 ms; echo time = 2.26 ms; flip angle = 9°; field of view = 240 × 240 mm; matrix size = 256 × 256; thickness = 1.0 mm; and 176 sagittal slices without interslice gap.

### 2.4. MRI Data Preprocessing

High-resolution T1-weighted data were preprocessed using the Computational Anatomy Toolbox (CAT12; www.neuro.uni-jena.de/cat/ accessed on 20 March 2022) and run through SPM12 (https://www.fil.ion.ucl.ac.uk/spm/software/spm12/ accessed on 21 March 2022) in MATLAB 8.4.0. (Mathworks Inc., Sherborn, MA, USA). The preprocessing steps involved spatial normalization to the same stereotactic space (using the Diffeomorphic Anatomical Registration Through Exponentiated Lie Algebra (DARTEL) algorithm implemented in SPM12) and segmentation (Figure 1). Modulated GM images were resliced to a 2 mm isotropic voxel size and spatially smoothed (Gaussian kernel of 6 mm full-width at half maximum).

### 2.5. Extraction of Individual Morphological Brain Networks

In the single-subject MBNs, nodes were defined as 90 anatomical regions according to the automated anatomical labeling (AAL) atlas, and edges were defined as the interregional similarity determined using the Multivariate Euclidean Distances (MEDs) approach [11,16]. In the kth subject, each pair of anatomical regions $\left(X,Y\right)$ from the AAL template was computed as a morphological similarity using the combined Euclidean distance $e_{k}\left(X,Y\right)$, which was defined as follows [14]:(1)$$e_{k}\left(X,Y\right)=\frac{n1n2}{n1+n2}\left(\frac{2}{n1n2}\overset{n1}{\underset{i=1}{\sum }}\overset{n2}{\underset{j=1}{\sum }}\|x_{i}-y_{i}\|{}_{2}-\frac{1}{n_{1}^{2}}\overset{n1}{\underset{i=1}{\sum }}\overset{n2}{\underset{j=1}{\sum }}\|x_{i}-y_{i}\|{}_{2}-\frac{1}{n_{2}^{2}}\overset{n1}{\underset{i=1}{\sum }}\overset{n2}{\underset{j=1}{\sum }}\|x_{i}-y_{i}\|{}_{2}\right)$$

Here, $X=\left\{x_{1},\;\ldots ,\;x_{n}\right\}$ and $Y=\left\{y_{1},\;\ldots ,\;y_{n}\right\}$, where x and y denote vertices in regions X and Y, respectively. In addition, $n_{1}$ and $n_{2}$ are the numbers of vertices in X and Y. The Euclidean distance is computed by the 2-norm ($∥.{∥}_{2}$). The distance $e_{k}\left(X,Y\right)$ will influence the morphological feature distribution; when intraregions have the same morphological feature, the combined Euclidean distance $e_{k}\left(X,Y\right)=0$. Then, min–max normalization between regions X and Y was performed to minimize possible bias in different ranges across different subjects. The similarity connectivity value of the kth subject was converted using the following equation:(2)$$c_{k}\left(X,Y\right)=\exp\left(-\frac{e_{k}\left(X,Y\right)-e_{k\_min}}{e_{k-max}-e_{k\_min}}\right)$$

Finally, a 90*90 symmetry similarity matrix of each subject was obtained. The values of the edges are circumscribed from 0 to 1, and 1 represents identical morphological feature distributions in the two AAL regions. The MEDs consider multiple morphological features of all of the vertices, and this meaningful and reliable method is beneficial for investigating brain abnormalities in neurological diseases [14].

### 2.6. Analyses of Network Properties

After constructing the individual similarity matrix, analyses of the network properties were performed using the Gretna toolbox (https://www.nitrc.org/projects/gretna accessed on 21 March 2022) based on the following steps [17]: (i) a wide range of sparsity (S) thresholds were set: 0.05<S<0.40 with an interval of 0.01; (ii) both global and nodal network properties at each sparsity level were calculated; (iii) the area under the curve (AUC) across the sparsity parameter S for each network metric was measured; (iv) the AUCs of global and nodal network properties were compared. Global topological properties included the clustering coefficient (Cp), characteristic path length (Lp), normalized clustering coefficient (γ), normalized characteristic path length (λ), small-worldness (σ), local efficiency (Eloc), and global efficiency (Eglob) [17]. Nodal topological properties included nodal efficiency, nodal degree, and nodal betweenness centrality [17].

### 2.7. Statistical Analysis

#### 2.7.1. Demographic and Clinical Variable Statistics

Differences in continuous demographic data between groups were assessed using Student *t* tests. Differences in categorical demographic data between groups were assessed by chi-square tests. The statistical analysis was performed using SPSS software (version 22.0; IBM Corp., Armonk, NY, USA).

#### 2.7.2. Network Topological Metric Statistics

To evaluate differences in network topological metrics between groups, the AUC of each topological metric was compared using a nonparametric permutation test. For multiple comparisons, we randomly reallocated all the values into two groups and recalculated the mean differences (repeated 1000 times), and the 95th percentiles of each distribution were used as the critical values for significance testing. The *p* values were corrected by the Benjamini–Hochberg false discovery rate (FDR) [18].

#### 2.7.3. Network-Based Statistics Analysis

The differences in pairwise connectivity were evaluated using the network-based statistics (NBS) approach (http://www.nitrc.org/projects/nbs/ accessed on 21 March 2022) as follows: (i) individual similarity matrices of the nodes with between-group differences in centrality measures (nodal degree, efficiency, and betweenness) were created, (ii) a set of suprathreshold links that included any connected components (*p* < 0.05) were defined, and (iii) the significance for each component with the nonparametric permutation approach (1000 permutations) was estimated [18,19].

#### 2.7.4. Correlation Analysis

For all significantly altered network metrics, a partial correlation analysis was conducted to examine the correlations with clinical variables, controlling for age and sex as confounding variables (FDR-q < 0.05; IBM SPSS Statistics V21.0).

## 3. Results

### 3.1. Demographic and Clinical Characteristics

Two patients and 1 HC with excessive head motion were ruled out, and 1 HC with unfinished scanning was ruled out. Finally, this study included 20 HCs and 23 young survivors of ALL (Table 1). No significant differences in age (*p* = 0.587) or sex (*p* = 0.319) were noted between groups. The young survivors of ALL had mean off-therapy durations that ranged from 12 to 18 months, lower IQ (78.35 ± 10.4), poor executive function (TMT-A, 94.17 ± 19.7), and worse verbal memory (DST, 4.13 ± 1.06).

### 3.2. Alterations in Global Morphological Network Properties

Based on all the defined connection densities, the young survivors of the ALL group and the HC group both showed a small-world architecture (γ > 1, λ ≈ 1, γ/λ > 1). The young survivors of ALL exhibited a lower clustering coefficient ($C_{p}$, *p* = 0.008) and local efficiency ($E_{loc}$, *p* = 0.035) than the HCs. No significant differences in the normalized clustering coefficient (γ), normalized characteristic path length (λ), small-worldness, or global efficiency were observed (Figure 2).

### 3.3. Alterations in Nodal Morphological Network Properties

Figure 3 and Table A1 list brain regions exhibiting significant between-group differences in at least one nodal metric in the young survivors of ALL compared to the HCs (FDR corrected, *p* < 0.05). Among the increased nodal centralities or efficiency, three nodes were located in the default mode network and in the auditory network, two nodes were located in the subcortical network, and other nodes were located in the frontoparietal, cingulo-opercular, and visual networks. Among the decreased nodal centralities or efficiency, two nodes were located in the subcortical network and in the visual network, and others were located in the default mode, frontoparietal, and salience networks.

### 3.4. Alterations in Connections Based on NBS Analyses

Using NBS, we identified morphological subnetworks in the young survivors of ALL showing hyperconnectivity and hypoconnectivity, including significantly increased connections involving 13 nodes and 6 edges and decreased connections involving 78 nodes and 46 edges, respectively (shown in Table A2), compared with the HCs (Figure 4).

### 3.5. Relationships between Network Properties and Clinical Variables

Off-therapy durations were correlated with nodal betweenness of the right superior occipital gyrus (SOG, *p* = 0.040) and nodal efficiency of the left middle occipital gyrus (MOG, *p* = 0.035). IQ scores were correlated with nodal betweenness of the right SOG (*p* = 0.026) and left angular gyrus (ANG, *p* = 0.024). DST scores were correlated with the nodal degree of the right Rolandic operculum (ROL, *p* = 0.048) and nodal efficiency of the right ROL (*p* = 0.023). TMT-A scores were correlated with the nodal betweenness of the right fusiform gyrus (FFG, *p* = 0.015).

In addition, several connection coefficients were correlated with off-therapy durations, DST scores, and TMT-A scores (Figure 5).

## 4. Discussion

We used structural MRI data to construct individual-based morphological networks to examine the topological organization of GM networks and their relationships to clinical variables in young survivors of ALL. There were two main findings: (1) at the global level, morphological networks showed decreased segregation reflected by lower Cp and Eloc and decreased nodal centralities and nodal efficiency in several nodes; (2) at the connection level, decreased morphological connections were mainly noted in the left and right pallidum (PAL) networks and increased connections were mainly noted in frontoparietal networks.

### 4.1. Decreased Segregation at the Global Level

In this study, the morphological networks in the young survivors of ALL showed decreased network segregation (reflected by lower $C_{p}$ and $E_{loc}$) thought to be relevant to information processing. However, the neurobiological meaning of individual-based morphological networks is not completely understood [12]. The possibility of selective neuronal injury by excitotoxic and apoptotic neurodegeneration of cytostatic drugs has been considered based on the administration of cancer chemotherapy during childhood to treat ALL, as these treatments induce GM density changes [20,21] that might impact the morphological similarity and network architecture. In fact, it has been widely reported that poorer overall functional status and even neurocognitive impairment occur in young survivors of ALL [2,22]. Specific GM and WM regions were positively correlated with memory and executive functions, including processing speed, working memory, and verbal learning [4,5,23]. However, in this study, no correlations were found between these altered global network properties and clinical parameters. These findings may indicate some degree of neural compensation or reorganization in response to structural injury in the young survivors of ALL. Therefore, a linear correlation was not observed.

### 4.2. Alterations in Nodal Centrality in Young Survivors of ALL

In this study, DST scores showed a trend toward a correlation with higher nodal centrality and efficiency in the right ROL. The ROL (Brodmann area 43) is a region with a variety of cytoarchitectonics and widespread connections and is involved in complex functions, such as sensory, motor, autonomic, cognitive, and language processing [24]. Specific GM and WM regions have a high cognitive reserve, allowing the efficient activation of these regions in specific tasks and playing a role in functional compensation following damage to these or adjacent brain regions. In addition, the off-therapy durations in young survivors of ALL were correlated with nodal centrality of the right SOG and nodal efficiency of the left MOG, suggesting that compensatory capacity gradually increased with the extension of time after the end of chemotherapy regimens.

We also found that reduced TMT-A performance may be associated with reduced nodal betweenness of the right FFG. The TMT-A has long been used to investigate deficits in central executive function. Previous studies have demonstrated deficits in central executive function in young survivors of ALL [25]. The FFG primarily includes structures involved in high-level vision. In ALL survivors, cognitive symptoms depend on central executive function; in addition, executive impairments are often found to be related to the disease itself and the toxicity of the treatment [23]. Taken together, the regions of decreased nodal centrality in ALL survivors are known to be involved in the processing of visual information. This finding suggests a reduction in overall information processing efficiency, which is consistent with the subtle versus pronounced profile of cognitive difficulties and learning delays observed in survivors of pediatric ALL.

We found two opposite associations between IQ and centrality at two increased nodes. A meta-analysis demonstrated clinically significant differences in the domain of intelligence in young survivors of ALL [26]. Chemotherapy damages DNA, either directly or indirectly, through an increase in oxidative stress that contributes to central nervous system (CNS) toxicity, as it is associated with leukoencephalopathy, resulting in neurocognitive deficits related to GM and WM atrophy. Our results do not explain the decrease in IQ, and more research is needed in the future.

### 4.3. Hyperconnectivity and Hypoconnectivity in Young Survivors of ALL

In this study, an interesting finding was that hypoconnectivity in the morphological subnetworks of the left and right PAL networks was associated with off-therapy durations. Wang et al. [10] reported that ALL patients showed decreased regional connectivity in a group-level structural covariance network when undergoing chemotherapy treatment. Longitudinal investigations also showed widespread reductions in brain architecture (i.e., GM volume, cortical thickness, and surface area) due to chemotherapy-induced apoptosis, reduced cell division, and impaired neurogenesis [3,4]. Our results are consistent with these findings and might add new evidence regarding the relationship of GM covariance with off-chemotherapy duration.

On the other hand, the PAL, which is a component of the striatum but also a transmission node connecting the prefrontal cortex (PFC) and amygdala, is involved in maintaining the muscular tone of the body and assisting in the control of movement. The TMT-A, which is part of the executive function test, was sensitive to impaired performance. Previous studies have revealed that neurocognitive performance (i.e., central executive functioning) was associated with structural and functional brain changes in ALL survivors [5,6,23]. Thus, our present findings suggest that decreased GM covariance might affect central executive function in young survivors of ALL.

In our study, the young survivors of ALL exhibited an increased morphological connectivity pattern, which involved the frontoparietal network. Using a functional connectivity approach, survivors of ALL showed patterns of hyperconnectivity in sensorimotor, visual, and auditory processing regions [7,8], whereas increased connectivity patterns have been shown to play a critical role in emotional and cognitive symptoms in ALL patients and young survivors of ALL [3,4,5,7,25]. Our findings are consistent with numerous studies showing increased centrality [6] and functional connectivity [8] in ALL survivors. These increases potentially reflect strengthened cerebral morphological integration in response to primary pathophysiological changes.

### 4.4. Limitations

This study has several limitations that should be noted. First, the number of participants recruited for this study was relatively small due to limited study funding and study time and other constraints such as the willingness of eligible children and parents to participate. In this context, the recruited ALL survivors were heterogeneous in terms of survival times. The sex ratios between patients and controls were also not entirely consistent, although they were not statistically significant. In future studies, the sample size could be expanded to include more in-depth studies with subjects with more consistent conditions. Second, the particular cognitive function scale that was used did not cover all aspects of neurocognitive function. This limitation may be one reason why the abnormalities identified in this study are not clinically relevant. In addition, pretreatment neurocognitive testing was not evaluated and documented. Third, we tested the reproducibility of our findings by constructing brain networks based on the Harvard–Oxford atlas templates and found similar results (Supporting Information: Appendix B). However, previous methodological studies have also shown that the results of graph theory studies are reproducible in different segmented brain templates [10,11,12].

## 5. Conclusions

Our study demonstrates that young survivors of ALL have a disrupted topological organization in MBNs, and these abnormal morphological connectivity patterns are apparent at the individual level. This reduced brain network efficiency may play a crucial role in the neurocognitive impairment of young survivors of ALL. Our future work will aim to also construct the individual-based functional brain networks and a comparative analysis between functional and morphological networks. Furthermore, a comprehensive assessment will be made of the changes in GM morphology and function of young survivors of ALL and their impact on neurocognition.

## Figures and Tables

**Figure 1 jcm-11-05362-f001:**
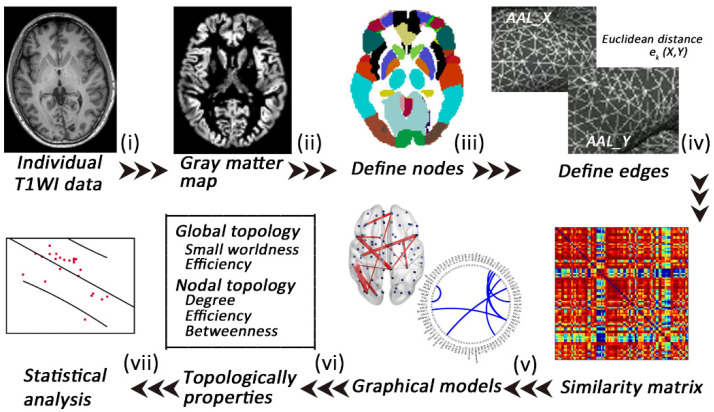
A flowchart of the construction of individual gray matter MBNs. (**i**) High-resolution T1-weighted imaging and (**ii**) data preprocessing (segment, normalize, modulate, and smooth); (**iii**) nodes are defined according to the automated anatomical labeling (AAL) atlases; (**iv**) edges are defined according to the combined Euclidean distance method; (**v**) an individual similarity matrix is obtained; (**vi**,**vii**) network properties are calculated and analyzed.

**Figure 2 jcm-11-05362-f002:**
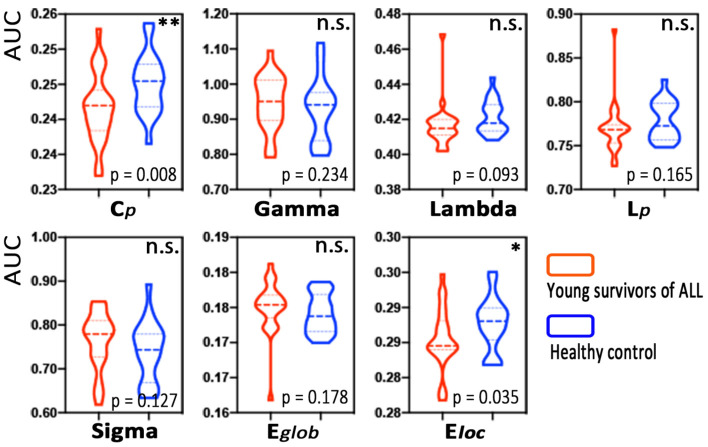
Group differences in global MBN properties between young survivors of acute lymphoblastic leukemia (ALL) and healthy controls. Note: * *p* < 0.05, ** *p* < 0.01, n.s., not significant.

**Figure 3 jcm-11-05362-f003:**
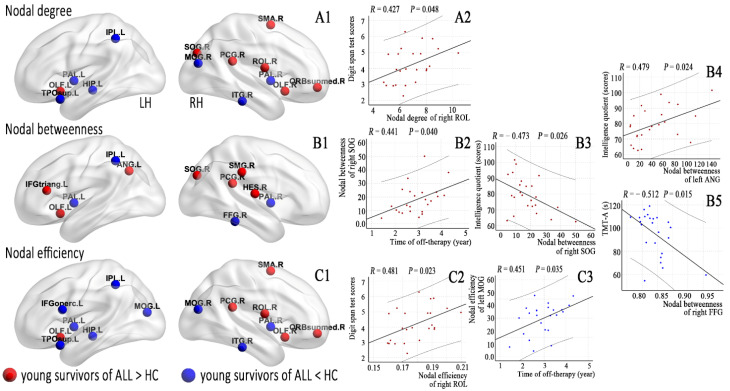
Brain regions with abnormal nodal centralities (**A1**,**B1**) and efficiency (**C1**) in the morphological brain networks were compared between the young survivors of ALL and controls (FDR corrected, *p* < 0.05) and (**A2**, nodal degree; **B1**–**B5**, nodal betweenness; **C2**,**C3**, nodal efficiency) correlated with the clinical variables in the young survivors of ALL.

**Figure 4 jcm-11-05362-f004:**
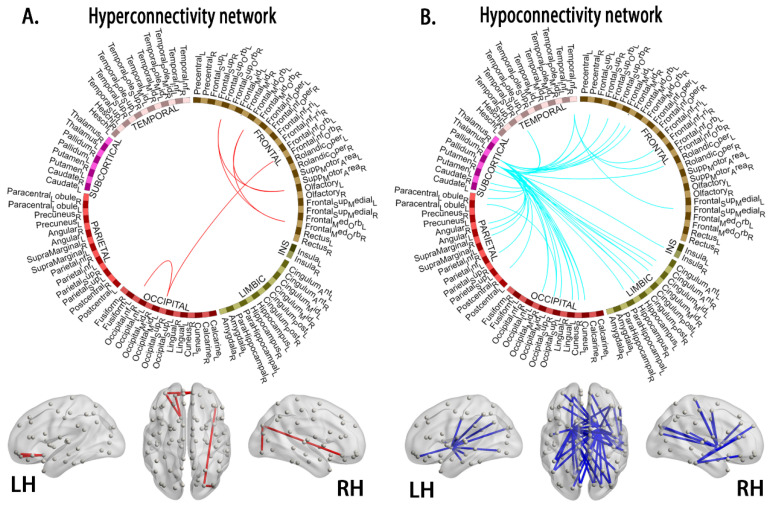
Compared to the healthy controls, the young survivors of ALL show network hyperconnectivity (**A**, interconnected links by red lines) and hypoconnectivity (**B**, blue lines). Note: ALL, acute lymphoblastic leukemia; LH, left hemisphere; RH, right hemisphere.

**Figure 5 jcm-11-05362-f005:**
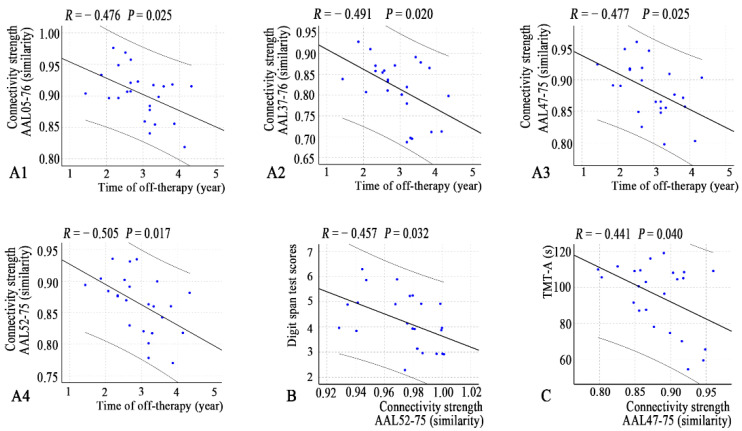
Correlations between connectivity strength and clinical variables (**A1**–**A4**, time of off-therapy; **B**, digit span test; **C**, TMT-A) in the young survivors of ALL. Note: AAL, anatomical automatic labeling atlas; ALL, acute lymphoblastic leukemia.

**Table 1 jcm-11-05362-t001:** Demographic and clinical characteristics of the young survivors of ALL and HCs.

	ALL Survivors	HCs	*p* Values
Sex (male/female)	16/7	10/10	0.319 ^a^
Handedness (right/left)	22/1	19/1	0.919 ^a^
Age (years)	10.3 ± 3.17 (6–16)	9.65 ± 2.28 (7–15)	0.587 ^b^
Off-therapy duration (years)	2.89 ± 1.43 (1–6)	n.a.	n.a.
Intelligence quotient (WAIS-IV)	78.35 ± 10.4	n.a.	n.a.
Trail Making Test A (s)	94.17 ± 19.7	n.a.	n.a.
Digit Span Test	4.13 ± 1.06	n.a.	n.a.

Abbreviations: ALL, acute lymphoblastic leukemia; HC, healthy control; n.a., not applicable; WAIS-IV, Wechsler Adult Intelligence Scale, Fourth Edition. Note: ^a^ Calculated by Wilcoxon rank sum test; ^b^ calculated by chi-square test.

## Data Availability

The data presented in this study are available on request from the corresponding author.

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
