# Peer review of "Is Brain Network Efficiency Reduced in Young Survivors of Acute Lymphoblastic Leukemia?—Evidence from Individual-Based Morphological Brain Network Analysis"

_jcm, 2022, doi:10.3390/jcm11185362_

Round 1

Reviewer 1 Report

Very interesting, nicely designed and presented manuscript.  Generally sections are concise, well written and easy to follow and data is very interesting especially as a potential pilot for future studies.

Study limitations are well described.  

questions/clarifications:

1. how were controls chosen?  Is there a reason why healthy siblings were not utilized as controls?  

2. How will this data be used for future study designs?

Author Response

Reviewer #1:

Very interesting, nicely designed and presented manuscript.  Generally sections are concise, well written and easy to follow and data is very interesting especially as a potential pilot for future studies.

Re: Special thanks to you for your comments.

Study limitations are well described.  

Re: Special thanks to you for your comments.

questions/clarifications:

  1. how were controls chosen?  Is there a reason why healthy siblings were not utilized as controls?  

Re: Line 107-111: “All of the HCs were recruited from the local area through community postings and screened using the Clinical Diagnostic Interview Nonpatient Version. HCs had no current or lifetime diagnosis of significant cognitive disorders, head trauma, or magnetic resonance imaging (MRI) contraindications.” was added to the manuscript.

Due to the current situation in China based on the family planning policy”, no siblings or very young, healthy siblings were not able to be recruited as a control group.

  1. How will this data be used for future study designs?

Re: Line 512-516: “Our future work will aim to also construct individual-based functional brain networks and a comparative analysis between functional and morphological networks. Furthermore, a comprehensive assessment will be made of the changes in GM morphology and function of young survivors of ALL and their impact on neurocognition.” was added to the revised manuscript.

Reviewer 2 Report

In present research work the authors have made an excellent effort to find out morphological changes in Grey matter in young ALL survivors and making a comparison with healthy ones. 

Overall the study is sound and interesting.

Kindly respond to following comments

Title:

1. since title has question mark at the end of first sentence, please make sure the word efficiency or efficient is most suitable.

2. Evidence from or evidence form? please double check 

Abstract: 

line 23: replace "in individual level" with "at individual level". 

line 28: remove " than the HCs. since comparison is mentioned mentioned already in line 26. 

Try to add more relevant key words mentioned in the research. 

line 57: change symptoms of patients to "symptoms of the patients". 

line 95: Rephrase sentence as " Clinical assessments were performed by two experienced pshychiatrists......

Results: 

Line 190: Please mention how many Young ALL survivors were approached or their files were extracted? 

Finally 23 young ALL survivors were included in the study who met the pre defined criteria. Health controls were selected by you right? 

Why did you select just 20 HC, why not 23 same as ALL patients? 

In HC 10 were male and 10 female. You would have selected 16 male and 7 female easily same as ALL patients. This would have made comparison of finding more homogenous. 

If there is any reason the mention it in a limitation part.  

Author Response

In present research work the authors have made an excellent effort to find out morphological changes in Grey matter in young ALL survivors and making a comparison with healthy ones. 

Overall the study is sound and interesting.

Kindly respond to following comments

Title:

  1. since title has question mark at the end of first sentence, please make sure the word efficiency or efficient is most suitable.

Re: Thanks for your comments, we have revised it as “Is brain network efficiency reduced in young survivors of acute lymphoblastic leukemia? --Evidence from...”.

  1. Evidence from or evidence form? please double check 

Re: We are very sorry for our incorrect writing, and corrected it in the revised version.

Abstract: 

line 23: replace "in individual level" with "at individual level". 

Re: Thanks for your comments, we have revised it follow your advice.

line 28: remove " than the HCs. since comparison is mentioned mentioned already in line 26. 

Re: Thanks for your comments, we have revised it follow your advice.

line 57: change symptoms of patients to "symptoms of the patients". 

Re: Thanks for your comments, we have revised it follow your advice.

line 95: Rephrase sentence as " Clinical assessments were performed by two experienced pshychiatrists......

Re: Thanks for your comments, we have revised it follow your advice.

Results: 

Line 190: Please mention how many Young ALL survivors were approached or their files were extracted? 

Re: We have added this information:” Two patients and 1 HC with excessive head motion were ruled out, and 1 HC with unfinished scanning were ruled out. Finally,...”.

Finally 23 young ALL survivors were included in the study who met the pre defined criteria. Health controls were selected by you right? 

Re:” All of the HCs were recruited from the local area through community postings and screened using the Clinical Diagnostic Interview Nonpatient Version. HCs had no current or lifetime diagnosis of significant cognitive disorders, head trauma, or magnetic reso-nance imaging (MRI) contraindications.”

Why did you select just 20 HC, why not 23 same as ALL patients? 

In HC 10 were male and 10 female. You would have selected 16 male and 7 female easily same as ALL patients. This would have made comparison of finding more homogenous. 

If there is any reason the mention it in a limitation part.

Re:”... First, the number of participants recruited for this study was relatively small due to limited study funding and study time and other constraints such as the willingness of eligible children and parents to participate. In this context, the recruited ALL survivors were heterogeneous in terms of survival times. The sex ratios between patients and controls were also not entirely consistent, although they were not statistically significant. In future studies...“.

Reviewer 3 Report

In this manuscript, the authors, Dr. Ying Zhuang et. al. recruited 23 young survivors of ALL and 20 healthy controls undergoing T1-weight MRI scanning. Individual-based morphological brain networks (MBNs) were constructed based on morphological similarity of gray matter using the combined Euclidean distance, after preprocessed and segmented. It’s very interesting that there are two main findings: 1. Morphological networks showed decreased segregation reflected by lower ?? and ???? and decreased nodal centralities and nodal efficiency in several nodes, at the global level; 2. At the connection level, decreased morphological connections mainly in the left and right pallidum networks and increased connections mainly in frontoparietal networks. Furthermore, the figures are very nice.

However, there are some minor issues need to be addressed.

1.       In the title, I think “evidence form individual-based…” should be “evidence from individual-based…”

2.       In the part of Abstract (page 1 line 26-27), suggest the authors delete the “Compare with the HCs,” because there is the comparison (than the HCs) at the end of this sentence.

3.       In Table 1 (line 196), “n/a ” should be “n.a.” to keep consistence with it in the table.

4.       In Figure 2, suggest the authors label the “no significant (n.s.)” on the images those have no statistical significance.

5.       In Table S2, please use three-line table, don’t need the vertical lines on the both sides.

6.       In Figure A1, can the authors make amend this figure to be high resolution, because the words around the circle are not clear.

Author Response

In this manuscript, the authors, Dr. Ying Zhuang et. al. recruited 23 young survivors of ALL and 20 healthy controls undergoing T1-weight MRI scanning. Individual-based morphological brain networks (MBNs) were constructed based on morphological similarity of gray matter using the combined Euclidean distance, after preprocessed and segmented. It’s very interesting that there are two main findings: 1. Morphological networks showed decreased segregation reflected by lower  and  and decreased nodal centralities and nodal efficiency in several nodes, at the global level; 2. At the connection level, decreased morphological connections mainly in the left and right pallidum networks and increased connections mainly in frontoparietal networks. Furthermore, the figures are very nice.

Re: Thanks for your comments.

However, there are some minor issues need to be addressed.

  1. In the title, I think “evidence form individual-based…” should be “evidence from individual-based…”

Re: We are very sorry for our incorrect expression. We corrected this phrase in the revised version.

  1. In the part of Abstract (page 1 line 26-27), suggest the authors delete the “Compare with the HCs,” because there is the comparison (than the HCs) at the end of this sentence.

Re: Line 27, The phrase Compare with the HCs, was deleted from the text.

  1. In Table 1 (line 196), “n/a ” should be “n.a.” to keep consistence with it in the table.

Re: In Table 1, “n/a” was corrected to “n.a.”

  1. In Figure 2, suggest the authors label the “no significant (n.s.)” on the images those have no statistical significance.

Re: Per the Reviewer’s suggestion, we have added the label “no significant (n.s.)” to the images in Figure 2. 

  1. In Table S2, please use three-line table, don’t need the vertical lines on the both sides.

Re: We have deleted “the vertical lines on both sides” in the revised version.

  1. In Figure A1, can the authors make amend this figure to be high resolution, because the words around the circle are not clear.

Re: We have provided a high-resolution image (900 dpi) of Figure A1 in the revised version.

Reviewer 4 Report

Please find the comments in the attachment.

Author Response

Major issues:

There are several limitations in the presented study. Recruited survivors were too heterogeneous in terms of treatment protocols and survival times. The used particular cognitive function scale did not cover all aspects of neurocognitive function. Although similar results were found, when the reproducibility of the findings by constructing brain networks based on the Harvard–Oxford atlas templates was tested, more templates are needed to test the reproducibility based on considerable variations.

Re: As the Reviewer suggested, we have added the following limitations to the revised version. First, the number of patients recruited for this study was relatively small due to limited study funding and study time and other constraints, such as the willingness of eligible sick children and parents to participate. In this context, the recruited ALL survivors.... “In future studies, the sample size could be expanded to include more in-depth studies with subjects with more consistent conditions. “In addition, pretreatment neurocognitive testing was not evaluated and documented. “However, previous methodological studies have also shown that the results of graph theory studies are reproducible in different segmented brain templates [10-12].

In total, the proposed conclusion “These results suggest a crucial role of decreased morphological connections of the PAL subnetwork with the prolongation of off-therapy duration and a relation to executive efficiency deficits in the young survivors of ALL” can not be derived from the presented data.

Re: We have rewritten this part according to the Reviewers suggestion as follows: This reduced brain network efficiency may play a crucial role in the neurocognitive impairment of young survivors of ALL.  

Please give more information about the treatment of the included patients (used treatment regimen, duration of treatment, age at treatment).

Re: We have added the following information to the text: “All patients were treated with a standardized three-year treatment using the Chinese Childhood Leukemia Group (CCLG)-ALL 2008 protocol [15].”

If data concerning neuro-cognitive testing prior to treatment is available, please present these data sets. If not, please state this.

Re: Unfortunately, there are no records of pretreatment neurocognitive testing. We state this information in the text as follows: “In addition, pretreatment neurocognitive testing was not evaluated and documented.”. 

Minor issues:

The whole manuscript text should be reviewed for minor spelling and punctuation issues (e.g. “morphological brain networks(MBNs)” (lines 60, 61))

Re: We apologize for our negligence of minor spelling and punctuation issues. We have carefully checked for these errors and fixed them.

Round 2

Reviewer 4 Report

I thank the authors for implementation of the required changes.